# Glial Cells as Therapeutic Approaches in Brain Ischemia-Reperfusion Injury

**DOI:** 10.3390/cells10071639

**Published:** 2021-06-30

**Authors:** Ivó H. Hernández, Mario Villa-González, Gerardo Martín, Manuel Soto, María José Pérez-Álvarez

**Affiliations:** 1Genomic Instability Group, Spanish National Cancer Research Centre (CNIO), 28029 Madrid, Spain; ihernandezh@cnio.es; 2Center for Molecular Biology “Severo Ochoa” (CBMSO) UAM/CSIC, 28049 Madrid, Spain; mario.villa@uam.es (M.V.-G.); msoto@cbm.csic.es (M.S.); 3Networking Research Center on Neurodegenerative Diseases (CIBERNED), Instituto de Salud Carlos III, 28031 Madrid, Spain; 4Departamento de Biología (Fisiología Animal), Facultad de Ciencias, Universidad Autónoma de Madrid, 28049 Madrid, Spain; gerardo.martin@estudiante.uam.es; 5Departamento de Biología Molecular, Facultad de Ciencias, Universidad Autónoma de Madrid, 28049 Madrid, Spain

**Keywords:** ischemic stroke, glia, neuroprotection, microglia, astrocytes, oligodendrocytes, therapy

## Abstract

Ischemic stroke is the second cause of mortality and the first cause of long-term disability constituting a serious socioeconomic burden worldwide. Approved treatments include thrombectomy and rtPA intravenous administration, which, despite their efficacy in some cases, are not suitable for a great proportion of patients. Glial cell-related therapies are progressively overcoming inefficient neuron-centered approaches in the preclinical phase. Exploiting the ability of microglia to naturally switch between detrimental and protective phenotypes represents a promising therapeutic treatment, in a similar way to what happens with astrocytes. However, the duality present in many of the roles of these cells upon ischemia poses a notorious difficulty in disentangling the precise pathways to target. Still, promoting M2/A2 microglia/astrocyte protective phenotypes and inhibiting M1/A1 neurotoxic profiles is globally rendering promising results in different in vivo models of stroke. On the other hand, described oligodendrogenesis after brain ischemia seems to be strictly beneficial, although these cells are the less studied players in the stroke paradigm and negative effects could be described for oligodendrocytes in the next years. Here, we review recent advances in understanding the precise role of mentioned glial cell types in the main pathological events of ischemic stroke, including inflammation, blood brain barrier integrity, excitotoxicity, reactive oxygen species management, metabolic support, and neurogenesis, among others, with a special attention to tested therapeutic approaches.

## 1. Introduction

Cerebrovascular accident or stroke is a major cause of long-term disability worldwide and the second leading cause of death [1,2,3]. The prevalence increases with advancing age in both males and females [1], but in some Asian countries, especially India and China, the prevalence of stroke in people under 40 years of age has recently increased, being a serious problem of public health [4]. The global tendency of an increment in the life expectancy predicts a parallel increase in the incidence of stroke in the next years. In Spain, according to data of Grupo de Estudio de Enfermedades Cerebrovasculares de la Sociedad Española de Neurología (GEECV-SEN), a stroke occurs every six minutes.

During 2020, a new risk factor for stroke occurrence has emerged related to COVID-19. Some reports indicated that COVID-19-related strokes are more severe and occur in younger patients than usual strokes, probably owing to coagulation abnormalities induced by SARS-CoV-2 infection [5,6]. The incidence of stroke in COVID-19 patients ranges from 1% to 6% [7].

Of all stroke types, the most common are ischemic (88%) and the rest are hemorrhagic (10% intracerebral and 2% subarachnoid) [1] (Figure 1B). Ischemic stroke (IS) occurs by a reduction or blockage of blood flow to the central nervous system (CNS) as a consequence of an embolus, thrombus, or atherosclerotic plate clogging a cerebral artery (Figure 1A). However, 30–40% of all ischemic strokes are cryptogenic, that is, of unknown cause [8]. The size of the initial damage area depends of the caliber of the affected artery. The brain tissue affected by ischemia is not homogeneously damaged. The region more severely hypoperfused, named ischemic core, undergoes mainly necrotic cell death, whereas the surrounding region, known as penumbra, is less affected and is characterized by a high rate of apoptotic cell death [9] (Figure 1A). Duration of occlusion is also an important factor that determines the severity of the damage and the prognosis of the patients.

Symptoms usually vary depending on the affected brain area and include sensory and motor dysfunctions that may be permanent. In fact, between 30 and 50% of stroke patients do not recover functional independence, which has an important socioeconomic impact [10]. To date, the only approved direct stroke treatments are intravenous thrombolysis with recombinant tissue plasminogen activator (rtPA) and endovascular thrombectomy. Although both treatments have been shown to be beneficial, the proportion of eligible patients is relatively low and there is a short time window from the onset of the symptoms for an effective treatment. Moreover, both strategies increase the risk of hemorrhagic transformation [11]. This has been the driving force of an increasing interest in understanding the cellular and molecular basis of IS as a way to find new treatments. It is already known that brain ischemia triggers a sequence of pathological events named as the “ischemic cascade”, which may endure from minutes to days [12]. These events include energy failure, excitotoxicity, oxidative damage, disruption of the blood brain barrier (BBB), inflammation, and finally cell death [13] (Figure 1C).

Research efforts focused on neurons as main players in IS have proved to render a very low rate of positive results. Instead, glial cells have recently made an entrance as promising therapeutic effectors and targets for the treatment of brain ischemia [14]. Here, we review the accumulated knowledge of glial cells role in the pathogenesis of IS with and special focus on treatment approaches.

## 2. Microglia

Microglia represent 10–15% of the total cells in the brain. These cells are considered as the resident macrophages and the first defense line in the central nervous system (CNS) against pathogens [15]. In physiological conditions, microglia have a high capacity to respond to changes in the CNS microenvironment owing to their processes [16]. These changes may be triggered by microorganisms such as bacteria or viruses, but also by neurodegenerative diseases such as Alzheimer’s disease, Parkinson’s disease, or cerebral ischemia. Particularly, microglia activation is one of the first events that occurs after an insult such as brain ischemia [17], taking place from minutes to a few hours after the start of the episode [18] (Figure 2).

### 2.1. Microglial Activation Timing upon Brain Ischemia

Morioka et al. [19], using Nissl staining, described an activation of microglia after 24 h of permanent middle cerebral artery occlusion (pMCAO) in cortical and thalamic regions. Years later, Schoroeter et al. [20] showed that, after 24–72 h of pMCAO, microglia presented star shape with thick and short processes located near the damaged area, while 6 days after pMCAO, microglia acquired an amoeboid shape in the same area. Thanks to the development of imaging techniques such as MRI, there has been deep progress in the study of microglia activation after cerebral ischemia. Rupalla et al. [21] described the activation of microglia as early as 30 min after pMCAO, showing hypertrophic cell body and processes in the penumbra. This revealed a high capacity of microglia to activate after a disruption in tissue homeostasis. Recently, a series of studies revealed the presence of activated microglia in the acute phase [22], sub-acute phase [23], and chronic phase [24] of IS. The aforementioned morphology changes are associated with different activated microglia functions. Some of these functions include an increase in phagocytosis rate, release of anti- or pro-inflammatory cytokines, proliferation, and migration [25].

### 2.2. M1 versus M2 Microglial Profiles

Under physiological conditions, microglia are in a “resting” status, from which they are able to lead a wide range of responses upon detection of changes in the environment like modulation of their dynamic processes, removing of debris and apoptotic cells by phagocytosis and remodeling of synapses [26]. Resting microglia present a low expression profile of surface molecular markers that include CD45, MHC-II, CD80, CD86, and CD11cc [27]. After an ischemic insult, activated microglia change this profile, presenting high expression of CD45, MHC-II, or CD86, among others. Moreover, Iba1, IB4, F4/80, and CD68 can also be used to identify activated microglia in this context [28]. Interestingly, there are differences in activated microglia molecular profile between the penumbra and ischemic core that could be used to delimit both regions. For example, the activated microglia in the penumbra are MHC-II+, associated with anterograde degeneration, while the core’s activated microglia are MHC-I+ phagocytic cells [29]. Activated microglia have been classically categorized into two general groups according to the paradigm of macrophage activation (Figure 2). In general, the scientific community named a pro-inflammatory profile as M1 activated microglia, while an anti-inflammatory profile was termed M2 activated microglia [30,31]. Despite the persistence of this binary classification, there are many works supporting a heterogeneity in the population of activated microglia and a coexistence of intermediate phenotypes as M2-a, M2-b, or M2-c [32,33,34]. Precisely identifying the different subpopulations of activated microglia that appear after an ischemic insult could be of great relevance for the development of effective treatments. Kanazawa et al. [35] defined a temporal polarization after ischemic stroke by activated microglia markers. They described a majority of M2 activated microglia population during the first 24 h after brain ischemia followed by an increase in M1 microglia population. Other studies support the idea that the balance between both activated phenotypes could determine the neurodegenerative disease progression, so that a majority of the M1 population is associated with a worse clinical prognosis [36]. Interestingly, activated microglia are known to switch from one profile to another and a great number of therapies are focused on this property [37,38,39]. M1 activated microglia contribute to an increase in the inflammation and cytotoxicity levels through the release of the cytokines TNF-α and IFN-γ; interleukins such as IL-1β, IL-6, IL-15, IL-18, and IL-23; chemokines like CCL2 and CXCL10; the metalloproteinases (MMPs) MMP-3 and MMP-9; and reactive oxygen/nitrogen species (ROS/RNS) [40,41]. On the other hand, M2 activated microglia promote the recovery of injured tissue and decrease inflammatory levels by secreting molecules such as IL-4, IL-10, IL-13, TGF-β, IGF-1, the neurotrophic factor BDNF, and vasoactive proteins [42].

### 2.3. Inductors of Microglial Activation in Stroke

There are several pathways that lead to microglial activation after brain ischemia. Classic activation is associated with M1 activated profile, while alternative activation is associated with the M2 activated profile [43]. There is a great range of molecular mechanisms underlying microglia activation after cerebral ischemia. Damage-associated molecular patterns (DAMPs) are molecules released in a passive way from cell debris or apoptotic cells after brain ischemia, driving microglial activation towards pro- or anti-inflammatory phenotypes [44]. Among these molecules, high mobility group box 1 (HMGB1) protein appears in the early stages of stroke and is recognized by several toll-like receptors (TLRs) like TLR2 or TLR4, which trigger an inflammatory response through the release of pro-inflammatory cytokines in an NF-κB-dependent process. In fact, TLR4 blockade has shown to be protective against brain ischemia with a reduction of the infarcted area, which could be due to a decrease in pro-inflammatory cytokines secreted by microglia [45]. Peroxiredoxins (Prdxs) constitute another important group of DAMPs with redox-activity. Prdx-1, Prdx-2, Prdx-5, and Prdx-6 are secreted by necrotic cells in the brain early after stroke inducing the release of pro-inflammatory cytokines, which are then recognized by TLR2 and TLR4 [46].

Initial disturbance of BBB integrity after an ischemic insult is also known to recruit and activate microglia, which start to secrete pro-inflammatory cytokines including IL-1β, TNF-α, and IL-6. IL-1β strongly induces activation of the astrocytes implicated in the neurovascular unit (NVU), promoting disruption of this functional structure, and thus leading to a metabolic uncoupling between neurons and the proximal blood flow [47]. The same population of activated microglia increases paracellular permeability of the surrounding blood vessels and further disrupts the NVU mainly through altered cytoskeletal organization, defective tight junction proteins’ (TJPs) expression, and MMPs’ release [37]. On the other hand, the M2 microglia population has been shown to exert angiogenic functions and promote BBB integrity, mainly through expression of TJPs [48].

Glutamate receptors (mGluRs) are also considered as microglial-related targets in brain ischemia. It is known that blockage or ablation of mGluR5 reduces acute microglial activation and promotes neuroprotection and neurofunctional recovery [49,50]. On the other hand, purinergic receptors, like P2X4R, P2X7R, and P2Y6R, have been related to neuroprotection mediated by microglia after brain ischemia owing to their anti-inflammatory effects [51,52]. This is a very complex process in view of the wide variety of receptors and channels on the microglial cell surface that get activated at the same time, triggering their own signaling cascade, and that must be tightly regulated to guarantee a harmonized response [53].

### 2.4. Therapies Based on Activated Microglia

From a therapy point of view, the available data suggest different alternatives to use activated microglia as a therapeutic target for brain ischemia. In vitro experiments using oxygen and glucose deprivation (OGD) offer an excellent model to study activated microglia response. These experiments usually focus on the molecular mechanisms underlying microglial polarization through the use of different agents such as LPS and IFN-γ, as a way to study the possibility to revert detrimental M1 profile and induce a neuroprotective M2 profile (Figure 2). On the other hand, in vivo experiments offer a better approach to study the role of activated microglia after an ischemic insult and their relation to other cell types, which remains poorly understood. By collecting data through these two approaches, we could know more specifically the response of activated microglia in this pathological context and develop effective strategies to reduce the detrimental effects of brain ischemia.

Minocycline is an antibiotic of the tetracycline family and it is the main compound used to abrogate inflammation induced by microglia activation [54]. Minocycline promotes a switch between microglial phenotypes, reducing the expression of M1 profile mRNA levels (IL-1β, IL-6, iNOS, and TNFα) and increasing typical M2 mRNAs (Arg-1, IL-10, TGF-β, and Ym1) [55]. Such an effect results in a reduction of the amoeboid morphology near the ischemic cortex and the infarct area [56]. Minocycline is also known to reduce cell death through the STAT1/STAT6 pathway [55]. Moreover, LPS-activated BV-2 microglial cell line presented a reduction in pro-inflammatory markers (CCL2, IL-6, and iNOS), with a concomitant decrease in caspase 3/7 activity and cell death [57]. Additionally, there are many studies supporting the idea that minocycline also increases the integrity of blood vessels, contributing to maintenance of the BBB integrity by an M2 microglia polarization [58,59].

Minocycline is one of the few glial-related components that have been tested in several clinical trials with promising results. An open-label, evaluator-blinded study of 152 patients showed minocycline, administered within 6 to 24 h of onset of stroke, to be associated with significantly lower National Institutes of Health Stroke Scale score and modified Rankin Score (mRS) compared with placebo [60]. By contrast, a multicenter randomized, double-blind, placebo controlled trial, “Neuroprotection With Minocycline Therapy for Acute Stroke Recovery Trial” (NeuMAST), in which patients were orally administered with either minocycline or placebo within 3 to 48 h of symptom onset, failed to prove any long-term beneficial effects over neurological outcome [61].

Apart from minocycline, there are a number of other compounds with promising efficiency in reverting microglia-derived detrimental effects in IS. Melatonin administration post-stroke is able to switch from M1 to M2 activated microglia through the STAT3 pathway, increasing secretion of anti-inflammatory cytokines and, therefore, reducing the damaged brain area [62]. Using GAPIs (a molecular cocktail from *Ginkgo biloba*), Zhou et al. [63] showed a reduction in the secretion of pro-inflammatory cytokines in BV-2 microglial cells, which acquired an M2 phenotype. Protocatechuic acid used after a cerebrovascular accident reverts M1-commited microglial cells and promotes expression of M2 markers [64]. Mechanistic target of rapamycin (mTOR) inhibitors like rapamycin are known to reduce the inflammatory microenvironment and infarct volume, decreasing the number of Iba1^+^ cells and the expression of M1 markers after pMCAO [65]. ABIN1, a NF-κB inhibitor, has also been shown to attenuate both microglia activation and the levels of pro-inflammatory cytokines after brain ischemia [66]. Interestingly, L-3-n-butylphthalide, an extract from seeds of *Apium graveolens Linn*, administered during 7 consecutive days after 45 min of pMCAO, improved the sensorimotor functions and reduced brain infarct volume, leading to an M2 microglia polarization [67].

Electroacupuncture (EA) is another neuroprotective strategy known to inhibit inflammation after brain ischemia, with many physical points available to apply this therapy including Baihui (GV20), Shuigou (GV26), Neiguan (PC6), Hegu (LI4), and Taichong (LR3), among others. There are some studies reporting the use of EA preconditioning to decrease pro-inflammatory effects triggered by brain ischemia. Using EA pre-treatment before pMCAO in rats, Liu et al. [68] showed a significant reduction of the infarct volume paralleled by functional motor recovery. From a molecular point of view, there is a reduction in the levels of different pro-inflammatory effectors such as TNF-α, IL-1β, and IL-6 in the damaged area and blood serum after I/R. Moreover, EA blocks the nuclear translocation of NF-κB (p65), preventing the expression of p38 MAPK and MyD88. In addition, another work explained that EA was able to induce a α7nAChR-dependent significant decrease of infarcted area and improve the neurological outcome [69].

## 3. Astrocytes

Astrocytes account for 50% of the human brain volume [70] and are normally classified into two mayor types according to morphological and spatial criteria: fibrous astrocytes in the white matter and protoplasmic astrocytes predominant in the grey matter [71]. The functional implications of each cell type in ischemic stroke are beyond the scope of this review, so the global term “astrocyte” will be used herein to encompass this complexity. Astrocytes have long been considered as a mere buffering system to sustain the correct functioning of the neuronal circuitry. However, in light of the evidence accumulated along the last decades, it becomes clear that astrocytes are active players in every physiological functions of the CNS as well as in pathological events following ischemic injury.

### 3.1. Excitotoxicity Modulation

Astrocytes play a major role in glutamate uptake from the surrounding neuronal synapsis and its posterior recycling into glutamine, which can then be reused by neurons as a substrate for glutamate synthesis. To this end, astrocytes present several glutamate transporters on their cell surface in direct contact with tripartite synapsis space, including the Na^+^-dependent transporters EAAT1 and EAAT2 (human gene names), also known as GLAST and GLT-1 (mouse gene names). It has been reported that the astrocyte-dependent glutamate buffering system becomes altered shortly after ischemia in several ways. These include epigenetic modulation of GLT-1 and GLAST promoters, resulting in lower gene expression; aberrant histone methylation giving rise to dysfunctional, but not increased, expression [72]; and S-Nitrosylation of GLT-1 with a concomitant reduction in its activity [73]. At a later stage, a decrease in ATP levels in astrocytes, mainly in the ischemic core, induces glutamate transporters reversal, further contributing to glutamate excitotoxicity and neuronal damage [74]. In vivo upregulation of GLT-1 using ceftriaxone [75] or by targeted overexpression with adeno-associated viral vectors [76] was neuroprotective and reduced brain damage. Similar results were obtained in cultured astrocytes subjected to OGD [77]. Carnosine was also proved to preserve GLT-1 activity in astrocytes after pMCAO, improving neurological function and decreasing infarct size [78]. A compound similar to ceftriaxone, sulbactam, was efficiently used to prevent hippocampal neuronal damage in a rat global brain ischemia model, while this effect was suppressed by either antisense knockdown or pharmacological inhibition of GLT-1 using dihydrokainate [79]. Furthermore, antisense knockdown of astrocytic GLT-1, but not EAAC1 neuronal glutamate transporter, increased neuronal damage induced in a transient focal cerebral ischemia (tMCAO) rat model [80], highlighting the special relevance of astrocytes in buffering ischemia-induced excitotoxicity. Nonetheless, astrocytes are also known to exacerbate excitotoxicity upon ischemia by releasing glutamate into the synapsis through volume-sensitive outwardly rectifying anion channels (VRACs) and connexin hemichannels. In fact, knocking out Swell1 (*Lrrc8a*), the only obligatory subunit of astrocytic VRACs, results in reduced brain damage and better neurological outcome upon I/R [81]. In accordance with this, pharmacological inhibition of VRAC using tamoxifen in MCAO-subjected rats reduced infarct area and improved neurological outcome [82]. Additionally, astrocytic P2X7 receptors (P2X7Rs) are also able to release glutamate upon ATP binding [83]. Given the increase in ATP extracellular concentration that takes place in the early phases of ischemic injury, P2X7Rs could be significantly contributing to excitotoxicity.

Connexin 43 (Cx43) is the predominant connexin in astrocytes and localizes to the cell surface where it conforms gap junctions and hemichannels. It has been described that, early after an ischemic insult (1–30 min), Cx43 gap junctions are phosphorylated by several kinases, like MAPK, PKC, pp60Src, and casein kinase 1δ, which triggers internalization of Cx43 hexamers. The remaining Cx43 hemichannels are dephosphorylated in a subsequent stage (after 60 min), increasing their opening probability and allowing harmful molecules into the extracellular space (ECS), like ATP and glutamate [84]. Leptin was found to suppress Cx43 rise after I/R reducing brain damage in a mouse model of MCAO, and it also blocked Cx43 hemichannels in cultured U87 cells [85]. Mimetic peptides are another way to block hemichannels. More precisely, Gap19 is a highly selective blocker that spares gap junctions at the initial moments of treatment (first 30 min), while effectively blocking hemichannels. At higher exposure times Gap19 slightly inhibits gap junctions. Suppression of Cx43 by Gap19 showed beneficial effects in MCAO mouse models [84].

### 3.2. Astrocyte-Neuron Metabolic Relationships in Stroke

Neurons heavily depend on glucose oxidative metabolism for their normal functioning, which makes them selectively vulnerable to hypoxia/hypoglycaemia [86]. For this reason, the role of astrocytes as metabolic supporters is key for neuronal survival during an ischemic insult. Neuronal energy demands can be mirrored by vascular regulation through astrocytic signaling pathways in what is known as neurovascular coupling. Released glutamate upon synaptic activity is bound by mGluR5, which then leads to a rise in intracellular Ca^2+^ concentration ([Ca^2+^]_i_) through PLCβ1 activation. This event triggers the release of arachidonic acid (AA) from the membrane. In the presence of low surrounding oxygen pressure, as happens upon ischemia, AA is preferentially transformed into PGE2 by COX-1, and is then exported inducing vasodilation and diffusion of oxygen and glucose from the nearby blood vessels into brain parenchyma [87].

Astrocytes provide neurons with lactate as a precursor for the tricarboxylic acid (TCA) cycle, in a proposed mechanism known as “astrocyte-neuron lactate shuttle hypothesis” [88], which remains controversial [89]. As previously stated, synaptic glutamate uptake by astrocytes triggers Na^+^-K^+^ ATPase, which in turn stimulates glycolysis and glycogenolysis to produce lactate [90]. Metabolism of lactate inside neurons implies that the pyruvate dehydrogenase complex (PDHC) is susceptible to inactivation through oxidative stress generated upon ischemic insults [91]. Energy depletion in an ischemic scenario leads to an increase in AMP levels and activation of AMPK, which in turn phosphorylates and thus inactivates acetyl-coA carboxylase, with the subsequent decrease in malonyl-CoA, a natural inhibitor of mitochondrial carnitine palmitoyltransferase I (CPT-I) [92]. Consequently, hypoxia/hypoglycaemia lead to increased activity of CPT-I and higher production of ketone bodies (KBs) through mitochondrial β-oxidation of free fatty acids (FAs) obtained from the bloodstream [91]. Astrocytes release KBs into the ECS, captured by neurons through MCTs and used as precursors for TCA cycle. Given that PDHC becomes partially inhibited in the presence of I/R-derived ROS, KBs become the most important source of energy over lactate after brain ischemia [93].

Exogenous KBs could constitute an interesting therapy for I/R injury. KBs increase mitochondria health and activity, reduce ROS and astrogliosis [94], and increase neurotrophin secretion (BDNF, bFGF) [95]. It has been recently reported that the axis SIRT3–FoxO3a–SOD2 becomes upregulated upon treatment with KBs, increasing mitochondria complex I activity and reducing protein oxidation, with a concomitant improvement in neurological outcome after an ischemic insult [96]. Adiponectin could also constitute a good therapy as it promotes oxidation of FAs and production of KBs through AMPK activation [97].

### 3.3. Oxidative Stress Management

Endogenous ROS in the CNS is generated by the mitochondrial electron transport chain and NADPH-oxidized pathway, while reactive nitrogen species (RNS) mainly proceed from L-arginine metabolism by nitric oxide synthase (NOS) [98]. Clearance methods can be classified into enzymatic and non-enzymatic. The former group includes Nrf2-controlled ones, catalases, SODs, and glutathione peroxidase. Non-enzymatic methods consist of molecules able to scavenge ROS/RNS including glutathione (GSH), bilirubin, uric acid, melatonin, and vitamins C and E. Another non-enzymatic method is the thioredoxin (Trx) system, where NADPH is used to reduce cysteine residues on Trx, making it a potent antioxidant [99].

Neurons are less efficient than astrocytes in dealing with oxidative stress owing to a continuous repression of Nrf2, which is a master regulator of redox genes including *GCLC*, glutathione reductase, *NQO-1*, and *HO-1* [100]. Increased levels of Nrf2 have been observed in I/R injury mainly in the penumbra, both in mouse and human [101]. It has recently been described that Nrf2 activation in astrocytes relies on glutamate binding to NMDA receptors (NMDARs), which suffer subunit composition changes in different models of ischemia [102]. GluN3A is known to increase in MCAO mice [103], rendering lower [Ca^2+^]_i_ elevations, which is expected given the inhibitory effect of GluN3A on NMDARs [104]. This could negatively impact GSH production and global antioxidant capacity of astrocytes.

One detrimental effect of ROS accumulation in astrocytes during ischemia is the activation of NLRP3 inflammasome. This process may depend on a two-step event (priming and activation) or on a single event (activation). Detection of ischemia-related DAMPs by TLRs can prime NLRP3 transcriptionally through NF-kB activation, which induces expression of NLRP3 and pro-inflammatory cytokines in a process partially dependent on mtROS [105]. Another way of NLRP3 non-transcriptional priming is through its deubiquitination by BRCC3, which can be triggered by mtROS and is a crucial step for NLRP3 activation [106]. ROS accumulation can directly activate NLRP3, promoting the release of TXNIP from Trx, to facilitate inflammasome polymerization [107].

Adiponectin (APN) is an adipose tissue-derived hormone released into the bloodstream that increases upon ischemia [108] and presents neuroprotective properties [109]. A very recent study proves that APNp, an APN-derived peptide able to cross the BBB, reduces ROS and NLRP3-mediated inflammation. APNp was shown to increase AMPK activation, Nrf2 nuclear translocation, and Trx1 levels [110]. Ascorbic acid is another molecule able to scavenge ROS directly. It is produced in astrocytes by GSH-mediated reduction and then transported into neurons [111]. Oral administration of nanocapsuled ascorbic acid has been shown to reduce ROS-mediated mitochondrial damage [112]. Peng et al. [113] recently described that DJ-1, which is an important antioxidant molecule mainly produced by reactive astrocytes, exerts a neuroprotective function upon ischemia through upregulation of Nrf2 and a concomitant increase in GSH levels. The AMPK-PGC-1α axis, which is induced upon ischemia owing to an increase in AMP levels, drives expression of *GCLM* specifically in astrocytes, and thus facilitates GSH synthesis. Accordingly, those AMP analogous molecules, like metformin and AICAR, improve neuroprotection and are good candidates for therapies [114]. Astroglia-specific ROS scavengers metallothionein(MT)-I and MT-II presented increased mRNA levels early after brain ischemia and deficient mice for these two proteins presented larger infarct sizes after ischemic injury compared with control mice [115].

Given the conspicuous relevance of ROS-mediated neurotoxicity in I/R, promoting the aforementioned astrocyte-related mechanisms to scavenge these toxic species represents a promising therapeutic approach in stroke, especially those Nrf2-centered strategies.

### 3.4. BBB Integrity and Edema

Astrocytes play a prominent role in the maturation and maintenance of the BBB by controlling water abundance, ion homeostasis, and other osmotically-active molecules (Figure 2). Astrocytes endfeet cover almost all the vessel surface stablishing closed contacts between them as the main glial component of the NVU. This structure strictly controls the diffusion of molecules into the brain parenchyma.

At early phases of ischemia, astrocytes become reactive and swell as a result of increased uptake of glutamate, K^+^, and lactate at the endfeet, but also due to Na^+^/K^+^ ATPase failure. Both these factors induce a change in morphology in astrocytes that cannot retain their normal functions and lose physical connections with endothelial cells (ECs) [71]. AQP4 is the main way in which water goes into astrocytes upon ischemia, resulting in the dysfunction of the endfeet, and deletion of this transporter improves the outcome after the insult, reducing swelling and edema [116]. Additionally, Na^+^/H^+^ exchanger isoform 1 (Nhe1) has been shown to be abnormally activated upon ischemia, which provokes an overload of intracellular Na^+^ and a concomitant astrocyte swelling [117]. Astrocytic-selective Nhe1 KO improves BBB integrity after tMCAO and reduces astrocyte activation, pro-inflammatory cytokine secretion, and hemispheric swelling, improving the neurological outcome [118].

Prompted by an ischemic insult, astrocytes secrete several factors with dual effects over permeability and integrity of the BBB. Some of those factors are classified here into negative effectors and positive effectors.

#### 3.4.1. Negative Effectors

VEGF is a factor that promotes vascular permeability through a downregulation of TJPs and angiogenesis. In different models of I/R, inhibition of VEGF signaling proved to be beneficial for BBB integrity maintenance, either by blocking VEGF with specific antibodies [119] or VEGFR2 with SU5416 or by genetic ablation [120].

Another group of negative effectors is MMPs, in charge of degrading TJPs and extracellular matrix in a physiological process that facilitates angiogenesis and BBB permeability. It has been described that reactive astrocytes produce and secrete MMP-2 and MMP-9 [121]. Several studies reported an increase in the levels and activity of MMP-2 and MMP-9 after an ischemic insult in animal models and cell lines (for a review, see Michinaga and Koyama, 2019). Zhang et al. [122] showed that MMP-2 and MMP-9 contributed to degradation of ZO-1 protein, leading to BBB disruption, while the MMPs’ inhibitor SB-3CT reduced BBB permeability. Another study showed beneficial effects of MMPs’ inhibitor BB-1101 administration after brain ischemia [123]. Genetic ablation of MMP-9 also proved to prevent proteolysis of BBB after ischemia [124].

Astrocytes can produce NO by iNOS upon several stimuli like IL-1β and ROS [125]. NO has a known negative impact over TJPs on the BBB [126]. In line with this, NOS inhibitor Nitro-L-arginine methyl ester (L-NAME) was able to prevent BBB disruption after focal ischemia [127]. It is also known that astrocyte-released glutamate can induce NO synthesis by eNOS through NMDAR activation in endothelial cells, resulting in vasodilation and higher permeability [128].

Astroglia are also an important source of endothelin 1 (ET-1), which promotes BBB permeability [129]. In fact, ET-1 overexpression in astrocytes aggravates neurological outcome of I/R in several animal models, leading to higher brain edema, BBB disruption, neurodegeneration, and mortality that can be partially corrected by ABT-627, an ETA receptor antagonist [130]. Other ET receptor antagonists were also proved to have beneficial effects, as was the case for S-0139 in I/R rats, which reduced BBB permeability, edema, and infarct size [131].

#### 3.4.2. Positive Effectors

Sonic Hedgehog (Shh) is a factor expressed mainly by astrocytes in the CNS, which drives expression of anti-apoptotic genes and promotes cell proliferation and progenitor self-renewal through activation of transcription factor Gli1a. It has been shown that astrocyte-released Shh promotes BBB formation and integrity through Shh receptors in ECs [132]. Recombinant Shh diminished BBB leakage after I/R by activating angipoietin-1, which promoted an increase in ZO-1 and occludin expression [133]. Shh signaling pathway was also protective against ECs’ apoptosis [134].

Astrocytes also increase their basal expression of RALDH2 in different brain inflammation paradigms, giving rise to higher retinoic acid levels, which was shown to be beneficial for BBB integrity [135]. Although not investigated yet, retinoic acid could be beneficial against I/R-induced BBB damage provided it is an inflammatory scenario.

Additionally, astrocyte-derived IGF-1 is known to exert a neuroprotective role against brain damage [136]. Actually, human IGF-1 gene transfer controlled by GFAP promoter showed to improve neurological outcome after I/R in aged rats [137]. Furthermore, IGF-1 reduced ECs’ apoptosis, BBB permeability, and infarct volume in rats after I/R [138].

Finally, to fight edema, astrocytes are also able to release osmotically active molecules like taurine upon ischemia, in what is called “regulatory volume decrease”. Release of taurine seems to be regulated by intermediate filaments-controlled channels, given that astrocytes from *GFAP^−/−^Vim^−/−^* mice showed reduced secretion of this factor [139].

### 3.5. Inflammation and Glial Scar Formation

Activated astrocytes are key players in the response to many different brain damages. This activation is mainly characterized by hypertrophy, increased proliferation, and newly acquired specific functions driven by gene expression regulation. This process is known as “reactive astrogliosis” [140,141].

Early after a brain stroke, there are several stimuli triggering astrocytes’ activation, like the release of neurotransmitters from nearby dying neurons, blood extravasation, hypoxia, and cell death, as well as release of cytokines from injured neurons and microglia, including TGF-α, CNTF, IL-1, IL-6, and KLK6 [71].

Within minutes after activation, reactive astrocytes produce and secrete many different pro-inflammatory molecules with detrimental effects over neuronal viability, like the cytokines IL-6, TNF-α, IL-1α, IL-1β, and IFN-γ, as well as ROS/RNS [142]. P2Y1 astrocytic receptors are known to induce release of pro-inflammatory cytokines and chemokines, aggravating brain damage after stroke [14]. In fact, it has been described that P2Y1 agonist treatment has a deleterious effect, increasing infarct area [143], while antagonists over P2Y1 have a positive effect on astrocyte viability upon ischemic damage [144]. TLR4 stimulation by ischemia-generated DAMPs on astrocytes’ cell surface induces secretion of IL-15, which acts as a chemoattractant for CD8+ T cells and natural killer cells, aggravating brain damage [145]. Consistently, antibodies directed against IL-15 improve the disease outcome [146]. On the other hand, astrocytic TGF-β cytokine has a neuroprotective function and its specific inhibition in astrocytes results in excess inflammation and higher damaged brain area [147]. This evidence highlights the dual role of reactive astrocytes upon ischemia. In fact, two main reactive astrocyte phenotypes have been proposed (Figure 2). A1 astrocytes are mainly induced by IL-1α, TNF-α, and C1q secreted by microglia and present a neurotoxic phenotype. By contrast, A2 phenotype is neuroprotective through neurotrophic factor release [140,148]. Ischemia-induced reactive astrocytes tend to neuroprotective phenotypes, as stated by RNA-seq data analysis of MCAO mice, which showed a predominance of A2-specific genes [149].

Days after an ischemic insult, a glial scar can be detected, displaying a Janus-faced role. Glial scar formation requires astrocyte multiplication in the penumbra and a posterior migration to the core border, where they secrete extracellular matrix proteins to avoid immune cells’ infiltration towards healthy tissue [150]. Astrocytes secrete chondroitin sulphate proteoglycans (CSPGs) as the main integrating factor of the glial scar with a well described role in axonal sprouting inhibition. Interestingly, cholinesterase ABC significantly reduces the inhibitory effect of CSPGs over axonal growth after brain ischemia in rats [151]. It has been described that *GFAP^−/−^Vim^−/−^* mice show less organized and less compact glial scar after brain injury [152] with larger infarct areas [153], suggesting a key role of intermediate filaments in proper astrocyte activation and scarring. On the contrary, knockout mice for CD36 astrocytic receptor, which mediates activation and glial scar formation, present 49% less infarct volume after an ischemic injury than their WT counterparts [154], and an attenuation of GFAP induction and glial scar formation [155]. It has been recently described that reactive astrocytes also acquire phagocytic activity in the penumbra of MCAO mice a few days after the insult, and this process depends on the upregulation of ABCA1, MEGF10, and GULP1 [156]. It is known that knockdown or knockout of any of the previously mentioned factors decreases astrocytes’ ability to phagocyte in vitro. Moreover, ABCA null mice present larger damaged brain area after ischemia [157], suggesting that the previously unsuspected role of astrocytes in debris clearance is key for ischemic injury resolution.

### 3.6. Neurogenesis and Synaptogenesis

Promoting neurogenesis is a promising therapeutic approach after an ischemic insult, where astrocytes, alike in other processes, play a dual role. As previously mentioned, glial scar is a good example of this behavior. It is known that glypican treatment, which is normally deposited by astrocytes in the peri-infarct region, reduced GFAP immunoreactivity and scar formation, but it also improved neurite outgrowth and behavior outcome [158]. In line with this, inhibition of the glial scar-forming enzymes chondroitin polymerizing factor and chondroitin synthase-1 by RNAi techniques improved neurite outgrowth [159].

Astrocytes secrete several factors able to differentially influence neurogenesis and plasticity after stroke. Ephrin-A5, which has a well-known inhibitory effect over axonal growth, is secreted by astrocytes in the peri-infarct zone. In fact, its inhibition promotes axonal outgrowth and functional recovery after stroke [160]. In contrast to this, astrocytes increase their basal expression of thrombospondins (Thbs1/2) upon ischemic injury, and those are essential for synaptic plasticity and functional recovery [161]. Additionally, it has been described that astrocytes secrete the chemokine stromal cell-derived factor-1 after MCAO, which functions as a neuroblast attractor to the ischemic lesion [162]. It was also shown that HMGB1, which is secreted by astrocytes upon ischemia, stimulates neural stem/progenitor cells differentiation into neurons in a PI3K/Akt-dependent manner [163].

Astrocyte-derived neurotrophic factors are crucial for neuronal survival after stroke. It is known that astrocyte-conditioned medium applied to MCAO brain reduces infarct volume [164]. BDNF is upregulated hours after stroke and its artificial overexpression in ischemic rat brain promotes neurite outgrowth [165]. Treatment with EPO also improves neurogenesis, brain remodeling, and neurorestoration after stroke in the perilesional zone and in the contralesional zone [166]. Activity-dependent neuroprotective protein (ADNP) [167] is a neurotrophic factor secreted by astrocytes with relevant roles in axonal transport stimulation, autophagy, neuronal sprouting, cell survival, learning, and memory [168]. An eight-amino acid ADNP-derived peptide named NAP, which is the minimal active peptide retaining neuroprotective properties, has been successfully used against brain stroke (MCAO) [169,170].

Interestingly, it has been recently described that striatal astrocytes can differentiate into neurons guided by a latent neurogenic program repressed by Notch signaling in basal conditions. Upon ischemia, Notch receptors 1, 2, and 3 along with their ligands DII1, Jagged1, and Jagged2 are downregulated, thus triggering astrocyte-dependent neurogenesis [171]. Direct reprogramming of glial scar astrocytes into neurons has also been achieved in brain-injured mice using viral-delivered NeuroD1 transcription factor [172].

### 3.7. Astrocytes as Preconditioning Vehicles

Ischemic tolerance (IT) induction or ischemic preconditioning (PC) consists of reducing the damage caused in a severe ischemic episode by provoking a previous mild ischemic insult and has recently appeared as an exciting therapeutic approach for I/R injury [173]. Astrocytic activation seems to be essential for induction of ischemic tolerance as previously preconditioned brains of GFAP^−/−^Vim^−/−^ mice show greater infarct volume upon ischemia compared with preconditioned brains of WT mice [153]. Hirayama et al. [174] showed that inhibition of initial microglial activation with minocycline did not prevent IT, while astrocyte inhibition with fluorocitrate showed no effective PC, suggesting that astrocytes are the relevant glial cell type in this process. They also showed that the axis P2X7R-HIF-1α is crucial for an effective IT induction. Moreover, it was described that a selective KO of neuronal HIF-1α does not prevent IT, indicating that the process does not depend on neuronal HIF-1α [175]. Another study has recently shown that exercise-mediated PC upregulates P2X7R and HIF-1α [176], reinforcing the role of these two factors in IT induction.

After a brief ischemic insult, reactive astrocytes increase glutamate transport, connexins, ZO-1, aquaporins, and neurotrophic factor release, all of which are supposed to mediate neuroprotection, but at the same time, ischemic PC is also protective for astrocytes, an effect that seems to be mediated by Nrf2 in primary astrocytes subjected to OGD [177]. These PC-treated astrocytes are more efficient, protecting neurons upon OGD possibly due in part to higher lactate release [178]. In line with this result, cross-tolerance mechanisms like epileptic PC seem to strongly depend on lactate transport into the mitochondria [179]. Other studies suggest that exosomes released by PC-treated astrocytes, which are then engulfed by neurons, could contain neuroprotective molecules such as miR-92b-3p [180].

Remote ischemic preconditioning (RIPreC), usually induced by blocking blood flow to the limbs, has repeatedly proved to induce tolerance against subsequent I/R damage in animal models and clinical trials [181]. In fact, it is known that patients with a history of peripheral vascular disease (PVD) present lower infarct volumes, better outcomes, and lower mortality rates after an ischemic insult [182]. Although there is no direct evidence supporting a role for astrocytes as the mediators of the mentioned protective effect, it could be interesting to test whether inhibition of astrocyte activation abrogates tolerance induction by RIPreC.

## 4. Oligodendrocytes

Oligodendrocytes are the responsible cells for axon myelination in the central nervous system. A correct myelination is crucial for correct nerve impulse transmission and electrical isolation of axons, but it also determines the axon diameter and helps in the maintenance of axonal stability, and thus neuronal viability [183,184,185].

In the adult brain, mature oligodendrocytes are spawned from oligodendrocyte progenitor cells (OPC) following a tightly coordinated process that includes migration, proliferation and differentiation. OPCs represent 5–8% of total adult brain glial cells and they are characterized by the expression of the neuron/glial antigen 2 (NG2) and platelet-derived growth factor receptor alpha (PDGFRA). They could play an important role in remyelination, as they show the ability to proliferate and differentiate after a demyelinating insult [186].

Pre-oligodendrocytes are the next maturation step and are characterized by expression of the cell surface markers O4 and O1 [187,188] and 2′,3′-cyclic-nucleotide 3′-phosphodiesterase (CNPase). At this differentiation stage, pre-oligodendrocytes connect with target axons, losing their bipolar form, and they start to build filamentous myelin outgrowths [184].

The main feature of mature oligodendrocytes is the production of mature and fully functional myelin, thus they are typically identified by the expression of myelin basic protein [189], transmembrane proteolipid protein [190], myelin associated glycoprotein [191], galactocerebroside [192], and myelin-oligodendrocyte glycoprotein [189].

Apart from myelin formation, there is evidence pointing to non-myelinating functions of oligodendrocytes, specifically of OPCs, that have shown certain immunomodulatory capacity and express cytokine receptors (Kirby et al., 2019). OPCs assess their environment through constant filopodia extensions [185] and, as microglia and astrocytes, they migrate to injured sites and acquire a pro-inflammatory phenotype that could negatively affect recovery after injury [193,194]. OPCs exposed to IFN-γ present high expression levels of MHC-I receptor and present antigens to cytotoxic T cells [194]. In addition, using in vitro and in vivo models of inflammatory demyelinating diseases, IFN-γ has been shown to inhibit differentiation into mature oligodendrocytes and myelination [195,196].

### 4.1. Oligodendrocyte Response to Ischemia

Most of the studies related to the effects of cerebral I/R in the CNS have focused on the gray matter, neglecting the effects in white matter. However, white matter damage is an important component to consider in ischemic stroke pathology, not only because of the direct negative effect of ischemia on this tissue [197,198], but also because of the indirect effect on the rest of the brain tissues given the potential role of white matter in tissue repair.

Brain ischemia causes severe damage to white matter, especially in the ischemic core. This white matter damage accounts for almost half of the infarct volume and is a major cause of functional disability and cognitive dysfunction [199,200]. Some recent reports suggest that changes in white matter infarct volume, particularly in the deep subcortical area, have clinical relevance as predictors of long-term severity after cerebral ischemia [201,202]. Interestingly, animal stroke models have revealed that the degree of white matter vulnerability to such insults strongly depends on age, with juvenile animals being more resistant to injury than perinatal or older animals [203,204], suggesting that different mechanisms of white matter injury are implicated in each developmental period.

Many reports show that preserving the integrity of white matter reduces neuronal injury and ameliorates neurological function [205,206]. Taking into account that disturbance of white mater is a direct reflect of oligodendrocytes perturbations, preserving their viability after stroke should enhance neurological recovery after brain ischemia.

It is known that oligodendrocytes are especially sensitive to ischemic damage in short-term periods owing to the high energy rate required for axon myelination, high intracellular iron levels, and low expression of certain antioxidant enzymes [207,208]. In the early stages of stroke, an increase in oxidative stress occurs, especially after reperfusion. This situation induces oligodendrocyte damage followed by tissue demyelination and thus axonal destabilization, affecting neuronal viability and promoting long-term neurological dysfunctions [209]. Oligodendrocytes exposed to hypoxia show a great production of superoxide radical, lipid peroxidation, and iron oxidation [207]. Using in vivo models of focal cerebral ischemia, it has been observed that, as early as 3 h after ischemia induction, oligodendrocytes start to show signs of swelling and can be lethally injured [198]. These changes precede neuronal injury by several hours, suggesting that white matter is more vulnerable to ischemic damage than gray matter at early phases of stroke. Besides, it has been demonstrated that NG2^+^ cells are more assailable than neurons or astrocytes during early reperfusion after 3 h of MCAO [210]. Other histopathological indicators of white matter damage after ischemia include segmental swelling of myelinated axons and the development of vacuoles between the myelin sheath and axolemma, indicating myelin destabilization [198,211]. It is known that antioxidants reduce ischemic damage. Administration of Ebselen right after reperfusion significantly reduced axonal and oligodendrocyte damage and improved neurological deficits in MCAO models [212].

Glutamate and ATP contribute to oligodendrocyte damage and injury of white matter [213]. As previously mentioned, extracellular levels of both neurotransmitters highly increase during ischemia above physiological levels, which triggers oligodendrocyte damage. Oligodendrocytes present functional AMPA and kainate glutamate receptor in their soma, while NMDA receptors can be detected in their processes [214,215]. The overactivation of these receptors enhances ischemic damage inflicted to oligodendrocytes through excitotoxicity [204,215]. On the contrary, glutamate receptor antagonists partially protect oligodendrocytes from ischemic injury and reduce white matter damage [216].

In parallel, ATP is released during ischemic events through the opening of pannexin-1 channels in a sufficient amount to activate low affinity receptors. Oligodendrocytes express low affinity P2X7Rs at relatively high levels [217]. ATP activation of P2X7Rs causes oligodendrocyte damage. Blocking these ATP receptors after MCAO has been shown to be protective against ischemic damage, not only preserving oligodendrocytes viability, but also maintaining the composition and functionality of axon initial segment, which promotes neuronal health in the injured area [218,219].

### 4.2. Oligodendrogenesis

Some studies highlight that cerebral ischemia induces an alternative long-term response of oligodendrocytes consistent in an increase of the cell population mainly at the ischemic area [220,221,222]. In other words, cerebral ischemia induces oligodendrogenesis, probably as a protective mechanism of self-repair in response to damage. As oligodendrocytes are damaged in the short term after ischemia, a successful replacement of damaged and lost oligodendrocytes with newly generated ones is essential for remyelination after brain injuries. Until very recently, the underlying mechanism and the origin of these new oligodendrocytes was poorly understood. These new oligodendrocytes may proceed from maturation/differentiation of a pool of precursors from two different origins: newly generated precursors from the subventricular zone (SVZ) or pre-existing NG2^+^ cells located in the gray matter that undergo differentiation. An increment of NG2^+^ cells in the penumbra region [186,203] in parallel with a significant reduction of these cells in the ischemic core after reperfusion has been observed in tMCAO models [223]. In addition, we recently reported a significant increment in a special subpopulation of oligodendrocytes (Olig2^+^ cells) expressing 3R-Tau in the SVZ at 5 and 21 days after pMCAO [220]. These data indicate that oligodendrogenesis exists after ischemia independently of reperfusion. We propose that some of these new oligodendrocytes could proceed from newly generated cells resulting from post-ischemic division in the SVZ during the first 6 h of stroke [220].

Therefore, ischemia induces cell division at SVZ and some of these newly generated cells become oligodendrocytes with the ability to migrate towards the damaged area, specially to peri-infarct regions; differentiate into mature oligodendrocytes; and promote partial tissue remyelination [220,223]. Although ischemic rats spontaneously recover neurological functions coinciding with an increment of Olig2^+^ cells in the ischemic area, neurodegenerative damage can still be detected long after the stroke episode. This suggests that neither the differentiation to mature oligodendrocytes nor the remyelination process are completely successful in restoring the histological structure, but they suffice for recovering certain neurological function [220,224].

Therefore, potentiating oligodendrocyte response to ischemic damage could be a good therapeutic strategy to ameliorate cognitive and motor disabilities induced by stroke. In line with this, treatments that promote oligodendrogenesis have been shown to enhance white matter repair and to reduce ischemic damage [206,225]. TGF-α increases after tMCAO in neurons and glial cells including oligodendrocytes. This effector directly protects oligodendrocytes from cell death induced by OGD, thereby maintaining white matter integrity and improving neurological recovery after stroke [205]. Accordingly, TGF-α-deficient mice showed long-term exacerbation of sensorimotor deficits after tMCAO accompanied by loss of white matter integrity [205]. Interestingly, administration of valproic acid, an antiepileptic drug, starting 24 h after MCAO, also increased oligodendrocyte survival and improved neurological outcome, associated with an increment in myelinated axons density in the penumbra [226].

In view of this evidence, we can conclude that cerebral ischemia induces oligodendrocyte death as a consequence of excitotoxicity and oxidative stress, while oligodendrogenesis is probably triggered as a defense mechanism to damage. These newly generated oligodendrocytes colonize penumbra area with a still unknown role.

Some authors suggest that these newly generated cells are in part NG2^+^ OPCs that never reach a mature state [227]. Alternatively, others authors propose that oligodendrogenesis could be a brain self-repair response triggered after ischemia in an attempt to myelinate injured axons or even promote neuronal survival by creating an optimal environment. Moreover, recent data demonstrated vascular effects of some newly generated oligodendrocytes. OPCs populations are heterogeneous and present different phenotypes, which give them multiples roles under pathological and non-pathological conditions. Apart from their regional variations, OPCs can classified by the base of their spatial relation to brain vasculature into perivascular, parenchymal, and intermediate. Perivascular ones are part of the so-called “neurovascular unit”. In a recent work, a subpopulation of newly generated OPCs after ischemia has been described, belonging to perivascular OPCs, that facilitates post-stroke angiogenesis, thereby improving functional recovery in a model of tMCAo [227]. The “vascular” effects of OPCs have been confirmed using OPCs’ transplantation in in vivo models of focal cerebral ischemia, which promotes integrity of the BBB by a reduction of leakage in the acute phase of ischemic stroke, alleviates edema, and improves neurological recovery after ischemic stroke [228]. These results point out an important vascular effect of newly generated OPCs after ischemia that would be interesting to enhance in order to improve tissue recovery through the functionality retrieval of the vascular unit.

Alternative approaches that stimulate OPCs’ division in adult brain after MCAO, such as bone marrow stromal cells’ (BMSCs) transplantation, showed improvement of remyelination [229,230]. Furthermore, using in vitro models has revealed that co-culture of BMSCs with oligodendrocytes increased oligodendrocyte protection by providing growth factors through activation of PI3K/Akt pathway. [231]. These results point out that cellular therapy based on transplantation of OPCs or BMSCs could be a new therapeutic approximation to cerebral ischemia focused in the restoration of white matter function.

Usually, myelin has been considered an immutable structure, but recent reports have revealed that myelin sheath can change throughout the lifespan, showing that myelination is not a static process [232]. Neuronal activity can affect proliferation and differentiation of oligodendrocytes [233]. Therefore, therapies aimed at increasing neuronal activity of the damaged area, such as physical exercise, improve the quality of life of patients and the overall health of their white matter. Physical exercise is known to induce newly generated OPCs or mature oligodendrocytes via activation of CREB/BDNF in a model of neonatal hypoxia-reperfusion [234]. In addition, exercise enhanced myelin repair by upregulating the Wnt/β-catenin signaling pathway and reduced infarct volume after brain ischemia in juvenile or adult rats [235].

### 4.3. Interaction with Other Glial Cells

As previously mentioned, ischemia induces neuroinflammation with a concomitant activation of microglia and astrocytes. Astrocyte-derived BDNF promotes not only neuron viability, but also oligodendrogenesis [236], and induces differentiation of OPCs into mature oligodendrocytes [237]. Conversely, in an in vivo model of ischemia, Sozmen et al. [238] showed that reactive astrocytes, probably belonging to the A1 subpopulation, blocked proliferation and differentiation of OPCs. This evidence supports the idea that A2-astrocytic phenotype, as opposed to A1, could reduce ischemic damage through an increase in functional oligodendrocytes number.

The relationship between microglia and oligodendrocytes after brain ischemia is similar to that observed for astrocytes. The population of microglia that colonizes ischemic area is able to induce a reduction of OPCs [232], but also to promote differentiation of OPCs [239]. This dual effect depends on the predominant microglial subpopulation in the damaged area, with the M2 phenotype being beneficial to oligodendrogenesis [240]. Minocycline administered after ischemia provides protection to white matter, attenuating OPCs and myelin loss in the neonatal rat brain, even one week after treatment [241].

## 5. Conclusions

Glial cells are progressively gaining interest in the field of ischemic stroke not only as mere supporting cells, but as determinant players in the development of subsequent I/R-associated neuropathologies [14]. This new perspective places microglia, astroglia, and oligodendrocytes as exciting therapeutic hotspots for the treatment of the second leading cause of death worldwide. However, the Janus-faced nature of glial cells upon ischemia imposes the need of a deeper understanding regarding the specific signaling pathways that trigger beneficial responses in each cell type.

Both microglia and astrocytes are frequently presented with bivalent phenotypes in response to an ischemic insult [30,140]. However, a wide variety of intermediate points between M1/A1 and M2/A2 extremes have already been identified, suggesting an even higher cellular and molecular complexity than initially anticipated. Moreover, the same cell type could still be playing dual roles in response to I/R as in the case of glial scar, where positive and detrimental effects have been observed upon interfering with scar formation. M1 phagocytic microglia is another example, where their detrimental effect over neuron viability may be counterbalanced by debris removal, which facilitates posterior neuronal sprouting. This evidence suggests that different phenotypes may be critical at specific disease stages and that neglecting *a priori* detrimental cell types could be imprudent. Such an intricacy will definitely have a negative impact on the possibility to translate research into clinical trials.

Oligodendrocytes are the newcomers in the field of brain ischemia and very little is known about the role of this cell type in this disease. However, and as opposed to microglia and astrocytes, oligodendrocytes seem to exert a net positive effect on neuron survival, which makes them an interesting target for therapeutic treatment, although a more comprehensive understanding of their specific role in white and grey matter is advisable. In any case, the fact that physical exercise stimulates OPCs’ production and maturation gives us a hint on the possible role of this cell type in the chronic phase of stroke, where neurological improvement has been observed upon rehabilitation programs [242].

In summary, the implication in BBB maintenance, inflammation, necrotic tissue removal, oxidative stress management, excitotoxicity, and neurogenesis, among others, represents a clear statement of the need to center clinical investigation on glial cells for the discovery of effective treatments in ischemic stroke.

## Figures and Tables

**Figure 1 cells-10-01639-f001:**
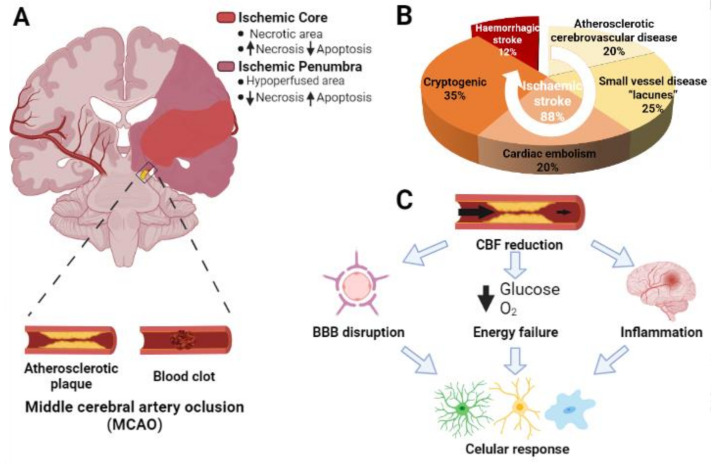
Main characteristics of the ischemic stroke. (**A**) Graphic representation of a coronal section of an adult human brain, which highlights the middle cerebral artery occlusion and its main causes: a blood clot and the formation of an arterosclerotic plaque. The ischemic core is highlighted in red and the penumbra region in fuchsia. (**B**) Epidemiological data of the incidence of the two main types of stroke (hemorrhagic and ischemic) and the main causes that lead to ischemic stroke. (**C**) Schematic representation of the main pathophysiological events triggered upon ischemia.

**Figure 2 cells-10-01639-f002:**
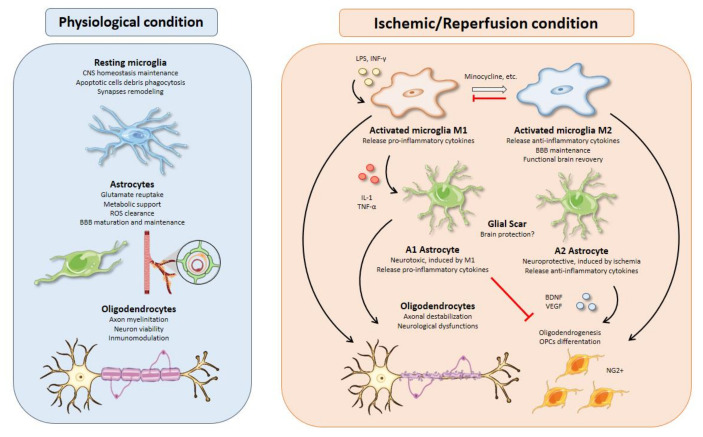
Glial cell functions, responses, and interactions in physiological and I/R conditions. **Microglial activation** is one of the earliest events after brain ischemia. The inflammatory landscape generated in the ischemic brain by inflammatory cytokines, debris, or molecules released from dead cells triggers microglia activation. Activated microglia are typically divided into M1 and M2 phenotypes. M1 microglia present a pro-inflammatory profile releasing cytokines such as IL-1 or TNF-α that can polarize astrocytes toward a neurotoxic phenotype, aggravating the inflammatory response. On the other hand, M2-microglia release anti-inflammatory cytokines, sustain blood brain barrier (BBB) integrity, and stimulate oligodendrogenesis through oligodendrocyte progenitor cells’ (OPCs) differentiation into NG2^+^ cells, thus promoting functional recovery after brain ischemia. All therapeutic strategies based on microglia are focused on minimizing the effects of M1-microglia using drugs like minocycline to switch phenotype from M1 to M2. Besides, M2 microglia could promote an inhibition of M1 microglia. **Activated astrocytes** are usually classified into A1 and A2 phenotypes. In line with M1 microglia, A1 astrocytes present a neurotoxic profile releasing pro-inflammatory cytokines with inhibitory effects over oligodendrogenesis and OPCs’ differentiation. In contrast, A2 astrocytes have neuroprotective functions releasing anti-inflammatory cytokines and trophic factors, such as BDNF or VEGF with similar effects over oligodendrocytes as those of M2 microglia. **Oligodendrocytes** are especially sensitive to oxidative stress and excitotoxicity generated during brain ischemia. Demyelination affects neurons owing to axonal destabilization, generating neurological dysfunctions. Moreover, these events are aggravated in the presence of M1 microglia and A1 astrocytes. At the same time, trophic factors released by M2 microglia and A2 astrocytes increase oligodendrogenesis and OPCs’ differentiation in order to repair damaged white matter in the injury area.

## Data Availability

Not applicable.

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
