# Peer review of "Glial Cells as Therapeutic Approaches in Brain Ischemia-Reperfusion Injury"

_cells, 2021, doi:10.3390/cells10071639_

Round 1

Reviewer 1 Report

Comments:

This manuscript titled “Glial Cells as Therapeutic Approaches in Brain Ischemia-Reperfusion injury” reviews the role of glial cells include astrocytes, microglia, oligodendrocytes, in the pathological events of cerebral ischemia-reperfusion injury. The manuscript is well organized, the content of the manuscript is informative, and the topic is the hot spot of current research on stroke. The manuscript will be improved by carefully editing.

Specific concerns:

The role of glial cells, including astrocytes, microglia, and oligodendrocytes in the neurovascular unit (NVU) after the cerebral ischemia-reperfusion injury, has rarely been described, although the author mentioned the neurovascular unit (NVU) at the part of astrocytes. The neurovascular unit (NVU) at the level of brain capillary is comprised of vascular cells (pericytes and endothelial cells), glia (astrocytes, oligodendrocytes, and microglia), and neurons. Cross talk between multiple cell types in CNS is very important for the homeostasis and normal function of the brain, and also remodeling the neurovascular unit after cerebral ischemia-reperfusion injury. The interaction of microglia or oligodendrocytes with neurons and endothelial cells after cerebral ischemia-reperfusion injury needs to be emphasized more in this review manuscript.

Reviewer 2 Report

This is a comprehensive review of glial cells in brain ischemia molecular pathways.The main focus on M1/M2 pheno/genotypes is interesting. Some literature on the role of lipid mediators could enhance this review (such as COX2 and PGs, and others). The references list is extensive and appropriate.

Some reference to human clinical trials (if available) would be appreciated.

I enjoyed reading this review. Some English editing is needed.
